

# Building a Raspberry Pi School Magnetometer Network in the UK

Ciarán D. Beggan[1] and Steve R. Marple[2]

[1]British Geological Survey, Research Ave South, Riccarton, Edinburgh, UK
[2]Physics Department, Lancaster University, Lancaster, UK

**Correspondence:** Ciarán Beggan (ciar@bgs.ac.uk)

**Abstract.** As computing and geophysical sensor components have become increasingly affordable over the past decade, it is now possible to design and build a cost-effective system for monitoring the Earth's natural magnetic field variations, in particular for space weather events. Modern fluxgate magnetometers are sensitive down to the sub-nanotesla level, which far exceeds the level of accuracy required to detect very small variations of the external magnetic field. When the popular

Raspberry Pi single-board computer is combined with a suitable digitiser it can be used as a low-cost data logger. We adapted off-the-shelf components to design a magnetometer system for schools and developed bespoke Python software to build a network of low-cost magnetometers across the UK. We describe the system and software and how it was deployed to schools around the UK. In addition, we show the results recorded by the systems from one of the largest geomagnetic storms of the current solar cycle.

*Copyright statement.*

## 1 Introduction

Over the past decade, the introduction of cheap reliable computing hardware and sensors, along with the ubiquity of high-speed Internet connections mean that it is now possible to make relatively robust and cheap magnetometer systems to accurately monitor the variation of the Earth's magnetic field over periods of several minutes to hours. This prompted us to investigate

the various types of sensors available and their ability to monitor the external magnetic field, and in particular their ability to record large geomagnetic storms.

The Raspberry Pi is a small single-board computer that typically runs the Linux operating system. Peripherals such as sensors may be directly connected to its general purpose input/output (GPIO) pins. Relatively cheap fluxgate magnetometer sensors are now available from a number of suppliers and analog-to-digital converters are also commonplace in the Raspberry

Pi ecosystem. For around one-hundredth of the price of a conventional scientific-standard fluxgate magnetometer instrument we found that we could build a system with approximately one-hundredth the accuracy and resolution. Despite the lower accuracy, this is still easily sufficient to record the daily variation of the Earth's ionosphere and magnetosphere, in order to detect the rapid changes in the magnetic field during geomagnetic storms. As a combination of a citizen science experiment and an outreach and education programme in geophysics, we applied to the UK Science Technology and Facilities Council





(STFC) in 2015 for a Small Public Engagement grant and were successful. In the application we bid for funding to build and deploy ten systems to schools across the United Kingdom.

In Section 2 of this paper we briefly describe the science of the magnetic field and what a sensor measurement consists of. In Sections 3 and 4 we outline how our system works and was deployed to create a school network. As an example of its utility

for science, in Section 5 we show the results of combining the Raspberry Pi magnetometer network with measurements from the permanent magnetic observatory network in the UK to enhance our understanding of the spatial and temporal dynamics of a large geomagnetic storm on the 7–8th September 2017. Finally, we discuss some of the lessons learned from the project in terms of public engagement and interacting with schools.

## 2   The Earth's magnetic field

The Earth's magnetic field is a vector quantity with a strength and direction, which varies both time and space (e.g. Merrill et al., 1996). It has an average strength on the surface of around 50,000 nT though this varies from 20,000 nT at the equator to 65,000 nT at the poles. Though the field is strong enough to move a small iron needle, it is incredibly weak; in reality the average fridge magnet is tens to hundreds of times stronger.

After temperature measurements, the magnetic field has one of the longest observational records available, dating back

centuries (Jackson et al., 2000). The vast majority (>95%) of magnetic field is generated in the liquid iron outer core by the self-exciting geodynamo. Similar to a bicycle dynamo, the Earth's core converts energy from motion of the conductive liquid iron into electrical currents. These electrical currents generate a long-lived magnetic field which is detectable on the surface of the Earth. In addition, remnant magnetisation of iron-bearing minerals in the upper crust act as another internal source but are small on average compared to the core field (typically <1 %). The remaining magnetic sources are external to the

Earth's surface and include currents flowing in the conductive ionosphere (at altitudes of 100 – 1000 km) and in the large scale magnetosphere created by the interaction of the Earth's magnetic field with the conductive solar wind. External fields are generally small (<4%) but can rise locally to over 10% of the field strength at high latitudes during large geomagnetic storms (e.g. Kivelson and Russell, 1995).

When a measurement of the magnetic field is made on or above the surface of the Earth, the value obtained is the sum of

all sources. Each source has a distinct temporal and spatial behaviour and, by measuring the field and its variation in many different places over time, the observations can be used to understand the individual geophysical systems. The crustal magnetic field changes on the slowest of time-scales – tens of millions of years on average. The core field changes on periods of around one year to millions of years, which includes global magnetic field reversals. The ionospheric magnetic field changes on a diurnal basis, driven by the effects of solar illumination and seasonally dependence on the solar elevation angle. Finally, the

magnetospheric field varies on time-scales of seconds to days.

Occasionally, when magnetic activity on the Sun's surface increases, for example from a Coronal Mass Ejection (CME), the speed and density of the solar wind (the tenuous ionised 'gas' which permeates the space between the planets) increases and energy is passed into the magnetosphere and ionosphere causing large electrical currents to flow. When this happens, intense





auroral electrojets form and move from their usual position near the poles and expand toward the equator, generating large magnetic fields which can be detected on the ground. The magnetosphere becomes loaded with energy which it attempts to dissipate every few hours through a process called magnetic reconnection. This causes the magnetic field to vibrate or pulsate at certain frequencies. These 'pulsations' last between a few seconds to tens of minutes and can be readily measured on the ground.

As most of the source signals overlap in time and space, they are difficult to identify with a single measurement. Hence, we need many sensors in a large network running for a long time in many different locations to resolve the contribution of each source. The global scientific magnetic observatory network fulfils this role. However, there are presently only around 200 magnetic observatories worldwide, with a very uneven spatial coverage biased towards the northern hemisphere, and Europe in particular (Love and Chulliat, 2013).

## 2.1 Magnetic field measurements

The geomagnetic field can be measured with an instrument generically known as a magnetometer. There are several types of instruments that can be used make a measurement such as (a) a simple compass needle to measure angles, (b) a copper wire wound around a cylinder of iron known as a fluxgate magnetometer to measure strength in a particular direction or (c) a magneto-resistive etching on a silicon chip (as in mobile electronic devices), again able to measure directional strength. The type of instrumentation used in scientific observatories has evolved over the past century from essentially compass needles suspended on quartz fibres requiring manual intervention to modern automated fluxgate magnetometers harnessing digital electronics.

For our systems we use fluxgate magnetometers which consist of a small cylinder of magnetisable iron wrapped by a copper wire with a large number of turnings (Primdahl, 1979). A electrical current, controlled by an oscillator, is passed back and forward through the copper wire creating a magnetic field. The magnetic field from the electrical current magnetises the core in one direction along its axis, then in the opposite direction. If a pre-existing magnetic field exists in the environment, such as the Earth's magnetic field, then it requires less current to magnetise the core along that direction. The current is directly proportional to the strength of the pre-existing magnetic field, meaning that fluxgate sensors can be calibrated to relate current to the magnetic field strength. Thus, a calibrated standard fluxgate sensor can measure the strength of the magnetic field along the direction of its axis and is sufficiently sensitive to the absolute level and variations down to the 1 Hz range of the field. The main drawback of fluxgates are their sensitivity to temperature change.

To measure the full magnetic field (as the magnetic field is a vector which has a strength and direction), three fluxgate coils set at right angles to each other are used. The convention (in geomagnetism) is to use an orthogonal Cartesian coordinate system (X, Y and Z) where the X-axis points toward geographical North in the horizontal plane, the Y-axis points to geographical East and the Z-axis points down towards the centre of the Earth. Thus, the strength and direction of a magnetic 'field line' passing through the sensor can be measured by how strong it is in each component. From the measurements of magnetic field strength made along each axis, a full set of all magnetic components can be calculated, including the declination (D) and inclination (I) angles and the strength of the horizontal (H) and total (F) field. Our system is a variometer which can only provide approximate



values of the strength and the relative change of the field, as compared to scientific observatories which make highly-calibrated absolute measurements of the geomagnetic vector and strength (Reda et al., 2011).

## 3  Instrumentation: development and build

The magnetometer system consists of (a) a sensor head which has three fluxgate magnetometers and (b) a Raspberry Pi com-
puter with an separate internal analogue-to-digital converter (ADC) board. The sensor head is linked to the ADC board via a connecting wire. Both the board and the magnetometers are powered by the Raspberry Pi. A blue LED within the sensor head indicates that the system is receiving power (Figure 1). The FLC100 magnetometers (from Stefan Mayer Instruments, Germany) are extremely sensitive to small variations of the magnetic field, with an operating range of zero to around 100,000 nT.

As noted, a standard magnetic field sensor has three magnetometers which are orientated along the three orthogonal components: North (X), East (Y) and Down (Z). The fluxgate coils are mounted in a Perspex block with a plastic base and brass screws (which are all non-magnetic).The sensor head also has a thermometer to measure background temperature. As it is so sensitive to temperature change, the system should ideally be kept at a constant temperature. A cover prevents accidental physical damage and helps reduce the rate of temperature changes to which the sensors are exposed.

The fluxgate magnetometers have been calibrated to output a 1 V analogue signal for a magnetic field strength of 50,000 nT. The analogue voltage is converted into a digital signal by the ADC. The 17-bit ADC is connected to the Raspberry Pi's Inter-Integrated Circuit ($I^2C$) bus using the GPIO pins. A cable connects the three fluxgate coils and the temperature to the ADC. The analogue signals are wired as single-ended inputs and so cannot make use of the 17th signed bit, giving the system a digitization resolution of $2^{16} = 65,536$ levels. Hence, the digitizer has a finite resolution which inherently limits the precision
of the magnetometer.

As an example, the smallest resolution at 50,000 nT/65,536 = 0.76 nT, meaning the digitizer can only resolve a magnetic field change larger than 0.76 nT. However, if the magnetic field strength is reduced to 15,000 nT, then the resolution increases i.e. 15,000 nT/65,536 = 0.22 nT. In practice, there are various factors that limit the resolution of the system so that the true accuracy is around 1.5 nT. This is sufficient to detect all the external field phenomena that are of interest to us.

To make a magnetic field measurement, the ADC samples the signals several times a second and then the Raspberry Pi averages the values measured over 5 or 10 seconds. Taking the average of a number of measurements has the effect of smoothing out the variation (from temperature changes, electronic noise and changes in the magnetic field) to give a mean value of the magnetic field over the short period. The Raspberry Pi is not fitted with a real-time clock and therefore it is important that it is connected to the internet in order to obtain and keep the correct time. The internet connection is also required for the transfer
of magnetic field data onto the web.



## 3.1 Operation and Testing

Our magnetometer system was designed to measure the external field variation rather than the core field. Although it also measures the strength and direction of the Earth's full magnetic field vector, the instrument is not stable enough over the long term (i.e. days to weeks) to be a useful instrument for studying the longer term field changes. As well as temperature,

the magnetometer is also sensitive to man-made magnetic fields from mobile phones, vehicles, or lifts in a building, for example. Consequently, we usually disregard the full field measurements and compute the variation of the Horizontal field ($H = \sqrt{X^2 + Y^2}$ around a quiet-time value, usually considered to be around 02:00–03:00 local time.

We tested an initial prototype in Edinburgh over several months in 2014 and compared it to the data from the closest geomagnetic observatory in Eskdalemuir, Scotland. Figure 2 shows an example of six days of data recorded on the Raspberry

Pi magnetometer in Edinburgh. The horizontal strength ($H$) was computed and the average value for the period was subtracted from the result to give the variation. The blue line shows the data from the Raspberry Pi. For comparison, the data recorded on a scientific instrument (called GDAS1) at the Eskdalemuir Observatory approximately 70 km to the south of Edinburgh is shown in black. The data from the Edinburgh system closely match the variation recorded at the observatory.

A number of different geomagnetic events were observed. On the 12th September there was a geomagnetic storm which

caused the magnetic field to fluctuate from +125 nT to -125 nT. The storm ended on the 13th September when the variations became smaller. The storm was followed on by a series of smaller rapid wiggles (pulsations) which are the result of dissipation of energy from the magnetosphere. By the 14th September, these disappeared and the ionospheric solar quiet time (Sq) current became visible. This is seen as a daily fall and rise of about 20 nT, with its lowest point at noon when the Sun passes overhead, on the 15th, 16th and 17th of September. Finally, on the 17th of September there is a step down and then up during the day

time. This is man-made, caused by someone changing the magnetic environment of the room (e.g. entering the room for a time and then leaving later in the day).

Further testing of the second iteration of the sensor design was carried out in late 2015. The updated magnetometer system was fitted with a solid-state temperature sensor and was deployed in Eskdalemuir for several weeks. It was placed in a non-temperature controlled building around 100 m from the GDAS1 system. Figure 3 (upper and middle panel) shows the

25 variation of the horizontal and vertical components as compared to the Eskdalemuir GDAS1 system. The lower panel shows the difference between the two systems with the temperature (scaled $\times 10$) plotted too. There is a strong correlation between the magnetic differences of the Raspberry Pi systems and the temperature changes recorded. The step during the 30th October is due to manual retrieval of the data, affecting the room temperature temporarily. However, though the systems are very temperature-dependent, it is possible to remove much of the error by 'backing-out' the measured temperature variation through

calibration.

Overall these tests indicate that, provided the Raspberry Pi magnetometer system is kept in a magnetically quiet and stable temperature environment, it is quite capable of capturing genuine geophysical phenomena, including the geomagnetic storms which we are most interested in.





## 4   Deploying the network

In order to promote the new magnetometers, we attended a number of professional development events run for secondary-level physics teachers to recruit potential schools around the UK. We also used contacts in the Institute of Physics and personal connections to find teachers who were interested in hosting the magnetometer at their school.

As noted, on its own a single magnetic sensor system is not particularly useful, but it can provide an educational tool for physics, astronomy, geology and geography students. However, tied into a UK-wide network of sensors, it is a means to participate in a genuine scientific collaboration to study the detailed variation of the magnetic field over the UK, particularly during geomagnetic storms. This is the 'pitch' we used to encourage teachers to host a system at their school. We provided an explanation of the science and tutorial notes for the teachers and pupils to read. We also suggested ideas of how to use the data, including how to process it using Jupyter notebooks and Python.

After our initial building and testing phases, we further developed the Python logging software to run the system automatically and pass the data back to the Internet. We sought to make the system as easy as possible to set up on site. To that end, the Raspberry Pi magnetometer software was set to start logging magnetic field data as soon as it is switched on. The system saves the outputs from the sensors to a file, containing the X, Y, Z magnetic field (in nT) and the temperature at the sensor head (in °C). The data are recorded every 10 seconds and placed into a comma separated value (.csv) file which can be read by Excel, Word or any other type of data processing software. The data are written to a particular directory on the Raspberry Pi which can be found under: `/data/bgs_sch/<site_name>/<Year>/<Month>/<site_name_YearMonthDay.csv>`

Two other files may also be created each day: a logging file (.log) and a bad data file (.csv.bad). The logging file has information about when the logging of data starts and ends. If there are problems with accessing the internet or time from a Network Time Protocol (NTP) server then data are written to the .csv.bad file instead to indicate there are uncertainties in the time record. Once every five minutes a Linux 'cron' job runs on the Raspberry Pi to transfer the recently recorded data to the Lancaster University website. Real-time plots of the data can be viewed at http://aurorawatch.lancs.ac.uk/plots/.

It took over a year to deploy most of the sensors. Figure 4 shows the final destinations of nine of the Raspberry Pi magnetometer systems across the UK. The aim was to cover the UK in both latitude and longitude and to complement the existing BGS and Lancaster networks. Benbecula in the Outer Hebrides is the most northerly and westerly point, Norwich is the most easterly point and Eastbourne is the most southerly point in the network. There is a cluster around Birmingham to tie in with a University of Birmingham cosmic ray experiment running at the Physics Department and in two local schools (Bordesley and King Edward's). Table 1 gives the locations of the each sensor. One of the systems was sent to the University of Otago in Dunedin, New Zealand as an experiment to show how the magnetic field varies simultaneously on a global basis during a geomagnetic storm. New Zealand is at the same geomagnetic latitude as the UK in the southern hemisphere.

The first system began running in October 2016 and expanded over 2016 and 2017. In the next section, we show an example of a major storm which was captured by the magnetometers.





## 5 Measurements from the geomagnetic storm of September 2017

In September 2017, one of the largest storms of the current solar cycle hit the Earth. A CME left the Sun at midday on 6th September and reached Earth's magnetosphere in around 36 hours. Starting around 23:30 UT on 7th September, the first and deepest part of the storm lasted for around 3 hours. Beautiful aurorae were visible all across the UK. Around 13:00 UT on 8th September a second burst arrived, though as it was during the day in the UK, the aurorae were not visible. The magnetic signature was detected by the Raspberry Pi magnetometer network. As the north (or X) direction is most sensitive to the auroral electrojet, we focus on this component in the plots shown.

To get a regional picture of the storm, data were collected from the Raspberry Pi magnetometers, as well as a number of other variometers and observatories around the UK, Ireland, Belgium, Germany and Norway. Figure 5 shows the location of the scientific observatories (INTERMAGNET), the BGS and Lancaster University AuroraWatch network of variometers (Case et al., 2017), a network in Ireland run by Trinity College Dublin (MagIE) and data from the Tromsø Geophysical Observatory (TGO) network.

The data were processed to remove the quiet time mean value of the X component at each site, using the value for 02:00–03:00 local time from the 7th September. Where the orientation of the sensor is unknown (as with the Raspberry Pi magnetometers) we rotated the horizontal components to match the estimated values of X and Y from a global magnetic field model (the International Geomagnetic Reference Field version 12). In Figure 6, we show the change of the magnetic field over the three days covering the 7–9th September 2017. The observatories and variometers are arranged by geographic latitude. The first and second parts of the storm are clearly visible as large spikes in the northern stations, decreasing in intensity further south. This is the signature of the auroral oval moving south during the peak of the storm, then returning northwards. In the first burst of the geomagnetic storm, the aurora moves as far south as LER and SUM which are in the Shetland Islands. The Raspberry Pi data (BGS$XX$) complement the other variometer and observatory data, matching the magnitude and timing of the main phases of the storm though they are slightly noisier overall. Sadly, not all system data were available at the time, so we did not completely cover the UK (e.g. Benbecula and Eastbourne are missing).

The line plot data was turned into a map of the magnetic field variation across the UK for each minute, interpolated using a physics-based method called Spherical Elementary Currents (Amm and Viljanen, 1999). Figure 7 gives a snapshot of the X and Y external magnetic field components at 23:45 UT as the auroral electrojet moves southward across the UK. It extends south of Shetland as can be observed most clearly in the X component (left panel). A video of the storm can be seen at https://youtu.be/ueDvVNhNbIc. In the movie, several more bursts of activity can be seen later in the day from 12:00 to 18:00 UT. Without the new magnetometers, it would not be possible to distinguish such detail. These measurements will lead to a better understanding about how the magnetic field changes over regions smaller than 1000 km during large geomagnetic storms at mid-latitudes.




## 6 Lessons learned

The idea of a school magnetometer network was in part inspired by the long-running UK Schools Seismometer network, part-run by BGS. Having experimented with the Raspberry Pi, it was an obvious and capable device for attaching all manner of sensors to. As with most projects, the concept and initial ideas of how it might work and operate differed strongly to the end
5  results. After making a prototype sensor and then bidding for a larger grant to make ten systems, we found that the building and development of the hardware and software was the (relatively) easy part – after all, this is what we do professionally.

However, once the systems were built, the harder parts of the project were the actual engagement with the schools and deploying the systems out to them. For half of the schools, we personally delivered the systems and helped to set them up. However, schools are busy places and it is difficult to get teacher's time and to hold their attention for more than a few
10  hours. More surprising was how tightly controlled school WiFi or Ethernet networks are. In fact much of the set-up time was spent attempting to gain access to an open internet connection within the school. Many of the schools were unable to continuously meet the magnetically quiet environment criterion, making stable measurements (without steps or temperature drift) over a single day hard to achieve during the working school week. Some of the schools managed to place the sensor in a quiet location but this reduces the opportunity for students to observe the system or to notice it on a daily basis. Finally,
once deployed, we found getting feedback from the teachers and schools was relatively slow. As the teachers rotate through different year classes after each year, the systems become unplugged or moved to noisier environments which further degrades their scientific usefulness. Hence, we monitor the data regularly and will contact the school to check if the data become poor or unusable.

Unfortunately, here is no obvious way to combat many of these issues — they are a function of the way schools operate.
However, we hope at least some students have been inspired to consider geophysics as a future career based on their interaction with the sensors and data.

## 7 Conclusions

We have developed a Raspberry Pi magnetometer system using standard off-the-shelf sensors and materials. The system costs around 100 times less than a scientific-grade sensor, but is sufficiently accurate to detect external magnetic field changes (at the
nanoTesla level) from geomagnetic storms. We describe the build, testing and deployment of systems to schools around the UK to create a countrywide network. We also offer an example of how the data can be used to supplement the existing network of scientific observatories and analyse a large geomagnetic storm on the 7–8th September 2017. This is an excellent example of how falling hardware costs in combination with science outreach can provide an interesting geophysical sensor network which can be used for both science and education.

*Code and data availability.* Data are freely available at http://aurorawatch.lancs.ac.uk/data/. Code for running the magnetometers is available at https://github.com/stevemarple/. Instructions for building the hardware can be obtained by contacting the corresponding author.



*Author contributions.* CDB built, integrated and tested much of the hardware. SRM developed the software to run the systems and transfer the magnetic data to the AuroraWatch UK website. Both contributed to the writing of the manuscript.

*Competing interests.* The authors declare no competing interests.

*Acknowledgements.* We wish to thank the STFC for awarding Small Public Engagement Grant in 2015 [ST/M006565/1] which made this
5 work possible. We acknowledge the support and collaboration of Prof Farideh Honary (Lancaster University) on this project. We thank the schools, teachers and technicians who helped to install and run the magnetometers on site. We also thank Tim Taylor, Tony Swan and Ted Harris at BGS for their support in building and testing the magnetometers before deployment and Sarah Reay for reviewing the manuscript. This paper is published with the permission of the Executive Director of the British Geological Survey (NERC).





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




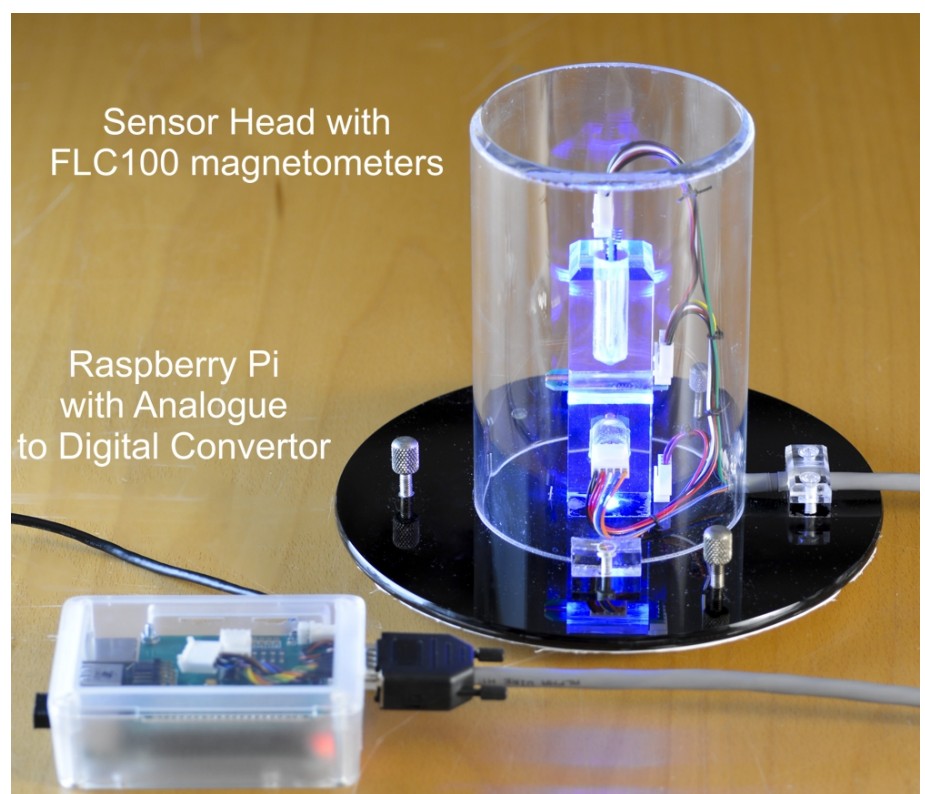

**Figure 1.** Raspberry Pi magnetometer system. The sensor head is approximately 15 cm high.

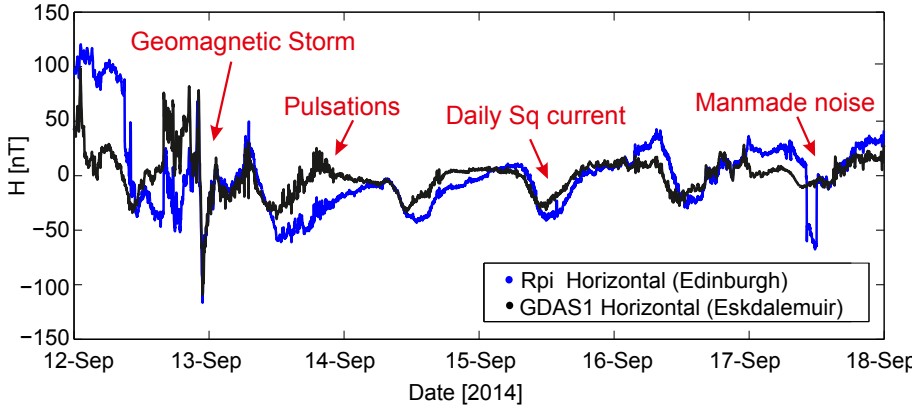

**Figure 2.** Horizontal field variation from 12-Sep-2014 to 18-Sep-2014. The blue line shows data from the Raspberry Pi magnetometer located in Edinburgh, UK. The black line is from the BGS GDAS1 system in the Eskdalemuir Geomagnetic Observatory (70 km south of Edinburgh).





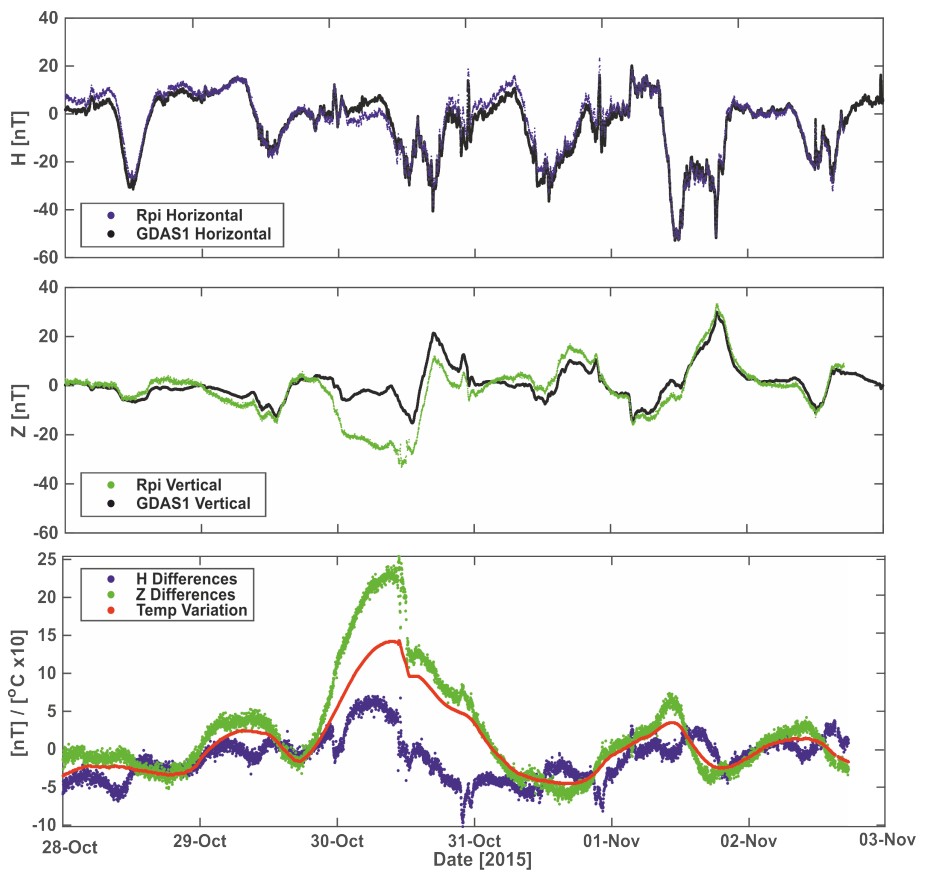

**Figure 3.** Comparison of the magnetic field variation between the Raspberry Pi magnetometer and Eskdalemuir GDAS1 system from 28-Oct-2015 to 03-Nov-2015. Upper panel: Horizontal component variation; Middle panel: Vertical component variation; Lower panel: Difference between the two systems with the temperature variation (x10) also plotted.

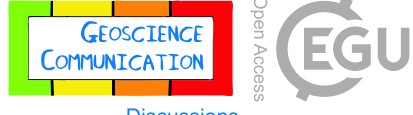
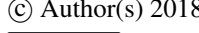


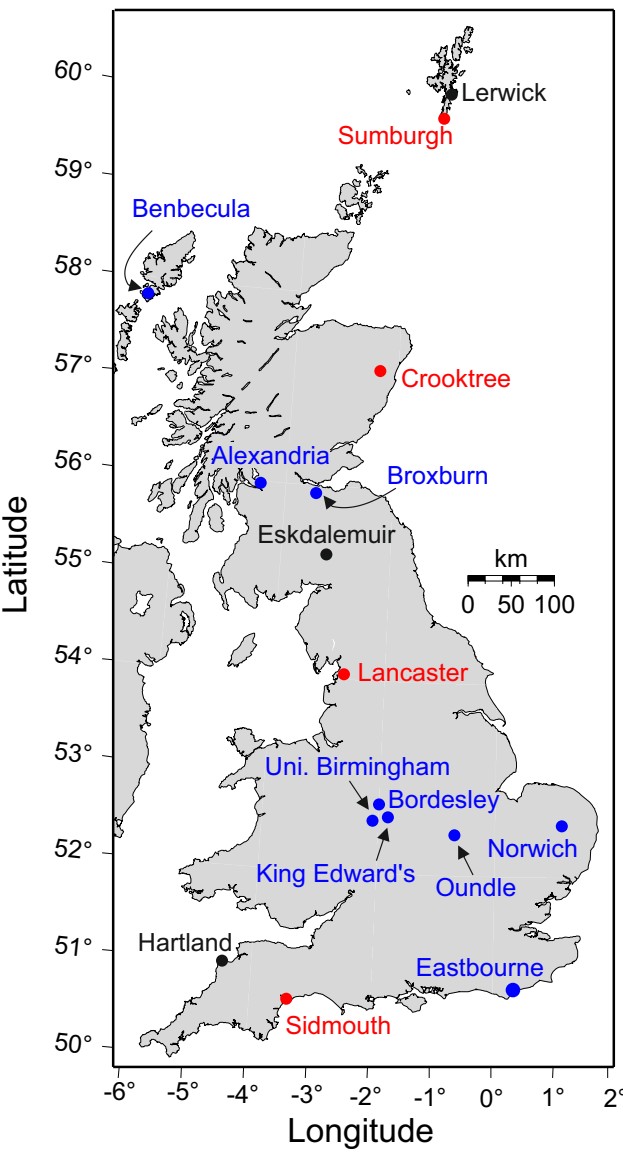

**Figure 4.** Map showing the locations of the Raspberry Pi magnetometers (blue), Lancaster University's AuroraWatch and SAMNET stations (red) and the BGS observatories (black).




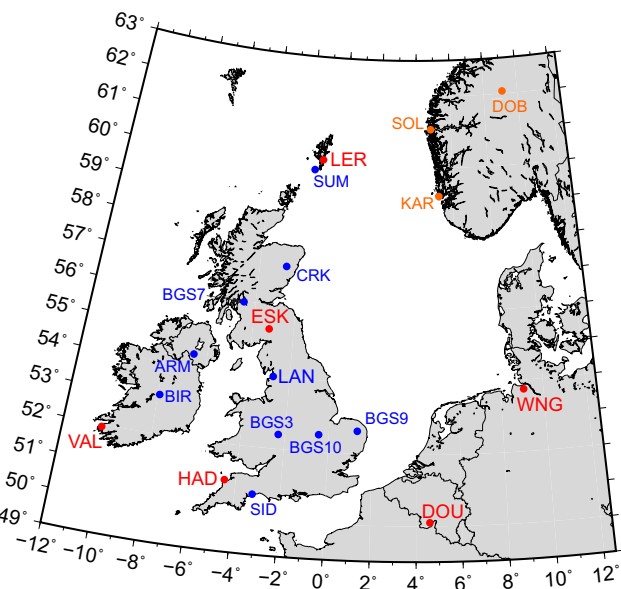

**Figure 5.** Map showing the locations of the variometer and observatory measurement sites available for the 7/8th September 2017. Red circles are INTERMAGNET observatories, orange circles indicated calibrated variometers (TGO) and blue indicate variometers including the Raspberry Pi (BGS) and the Lancaster AuroraWatch/SAMNET magnetometers.

**Table 1.** Identification number (c.f. Figures 5 and 6) and location of the ten Raspberry Pi systems.

| ID | Latitude | Longitude | Location |
|-------|----------|-----------|----------------------------------|
| BGS1 | 50.768 | 0.267 | Gilredge, Eastbourne |
| BGS2 | -45.80 | 170.50 | University of Otago, New Zealand |
| BGS3 | 52.450 | 1.929 | University of Birmingham |
| BGS4 | 52.451 | -1.925 | King Edward's School, Birmingham |
| BGS5 | 52.477 | -1.857 | Bordesley Green, Birmingham |
| BGS6 | 57.426 | -7.360 | Sgoil Lionard, Benbecula |
| BGS7 | 55.980 | -4.583 | Vale of Leven, Alexandria |
| BGS8 | 55.940 | -3.470 | Kirkhill, Broxburn |
| BGS9 | 52.632 | 1.300 | Norwich School, Norwich |
| BGS10 | 52.481 | -0.469 | Oundle School, Peterborough |




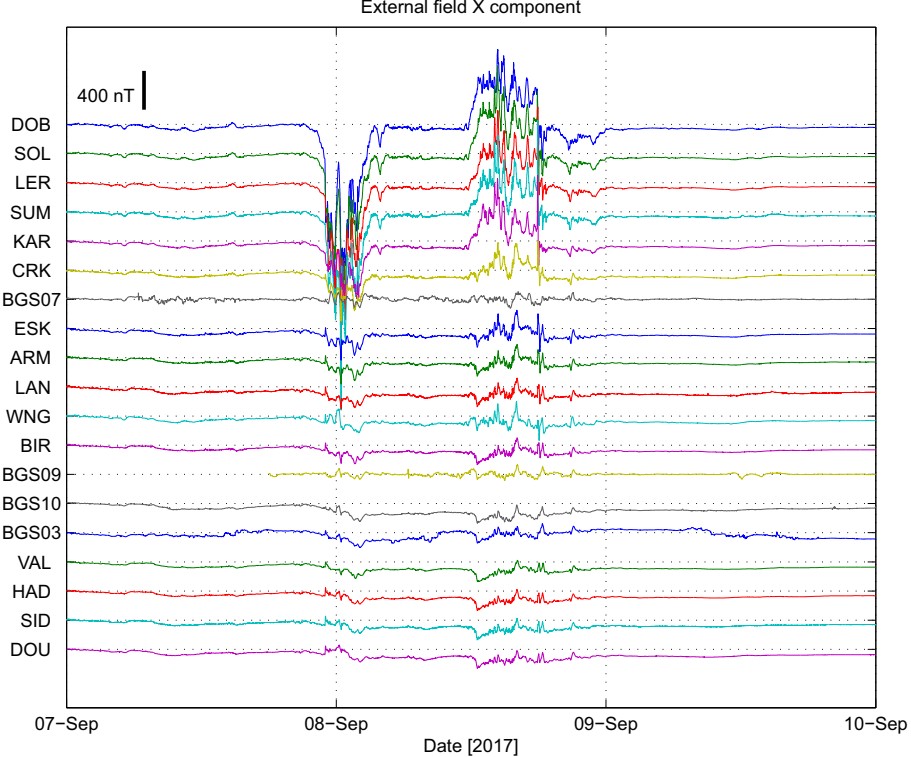

**Figure 6.** Stack plot of the variation of the X component over time at each measurement site from 00:00 UT 7th September to 00:00UT 10th September. Stations are ordered by geographic latitude. Note the initiation of the storm at midnight of the 7/8th September and the second phase beginning at 12:00UT on the 8th September. Scale bar denotes 400 nT change.

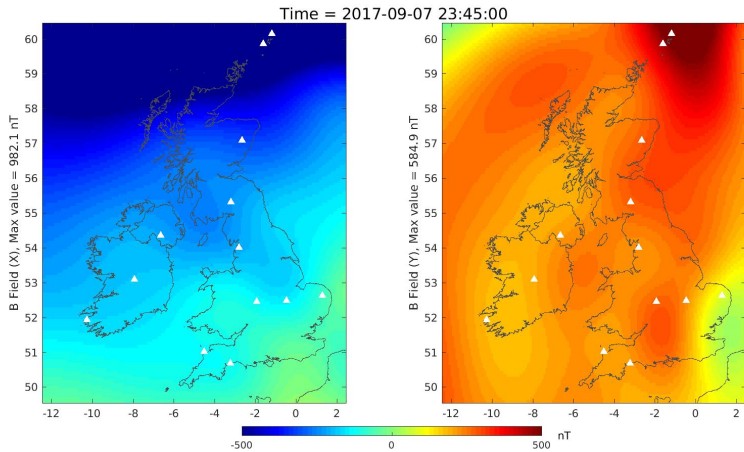

**Figure 7.** Snapshot of the two horizontal components of the magnetic field interpolated across the UK at 23:45 UT on the 7th September 2017.