# Peer review of "Building a Raspberry Pi School Magnetometer Network in the UK"

_Geoscience Communication, 2018_

## Referee Comment (RC1) · P. Livermore (Referee) · 16 Jul 2018

This paper presents a novel UK network of magnetometers that promotes engagement of schools with geophysics. The paper is suitable for GC, well written and interesting, illustrated with useful figures, and publishable in almost it its present form with a few minor technical corrections.

Technical typos 1. Abstract - last sentence, system not systems. 2. Introduction - "over periods" is ambiguous. Does this convey the time period i.e. 2 Pi/frequency or length of data window? 3. What is the actual price of one of the authors' magnetometers? Only a relative value is given. This might be helpful for others to see how much it would cost to join their network. 4. Section 2, l1 varies both  time.. 5. Section

2, l14, temperature measurements of what? Please be more specific. Presumably you mean of the atmosphere, and direct measurements rather than by proxy? 6. l29 typo: seasonal 7. Section 3.1, l23 the current <difference> is .. 8. Section 2.1, l33 can be calculated, <which may also be expressed as> D, I... 9. Section 3, l22 if the <calibration> magnetic field.. 10.Section 5. l4. Please make clear that X is the most sensitive component as the electrojet flows East-West. 11. Fig3. I found the bottom axis label confusing - it looks like its in units of nT/C. I suggest using a right axis, separating out the two quantities: i.e nT on the left axis, degrees C on the right. Also, you report only temperature variation - what is the baseline that you've used? i.e. what does zero temperature represent?

---

## Editor Comment (EC1) · M. Joubert (Editor) · 17 Jul 2018

Dear Phil, thank you for your concise and constructive review of this article. Much appreciated. Marina Joubert, technical editor, Geoscience Communication
* * *

---

## Author Comment (AC1) · 18 Jul 2018

Author's Response: Phil: Thank you for taking the time to review the manuscript. We have updated the manuscript as suggested and altered Figure 3 to be clearer. The detailed response is below.

1. Abstract - last sentence, system not systems. Response: changed

2. Introduction - "over periods" is ambiguous. Does this convey the time period i.e. 2 Pi/frequency or length of data window? Response: we meant the length of the data window over a time range of several minutes to several hours. Wording has been changed.

3. What is the actual price of one of the authors' magnetometers? Only a relative value
is given. This might be helpful for others to see how much it would cost to join their network. Response: The approximate cost is around 150 GBP at 2018 prices, though this is with a bulk-buy discount.

4. Section 2, l1 varies both  time.. Response: corrected

5. Section2, l14, temperature measurements of what? Please be more specific. Presumably you mean of the atmosphere, and direct measurements rather than by proxy? Response: It measures the ambient temperature of the air using a thermistor. The text has been updated.

6. l29 typo: seasonal Response: corrected

7. Section 3.1, l23 the current <difference> is .. Response: corrected

8. Section 2.1, l33 can be calculated, <which may also be expressed as> D, I... Response: changed

9. Section 3, l22 if the <calibration> magnetic field.. Response: I've left this as written because the resolution of each measurement is directly dependent on the field strength. Larger field strengths have lower resolution due to the limited range of the digisiter. In scientific observatories, the digitiser is 24-bit (rather than 16-bit) and the main field is 'backed off' using another set of coils wrapped around the fluxgate magnetometers to reduce the magnetic field variation to +/- 4000 nT for this very reason. These systems can achieve resolution of around 0.1pT or better, though other noise is around 5-10pT in general.

10.Section 5. l4. Please make clear that X is the most sensitive component as the electrojet flows East-West. Response: I've amended the text.

11. Fig3. I found the bottom axis label confusing - it looks like its in units of nT/C. I suggest using a right axis, separating out the two quantities: i.e nT on the left axis, degrees C on the right. Also, you report only temperature variation - what is the baseline that you've used? i.e. what does zero temperature represent? Response: The

Figure 3 plot has been changed to move the temperature variation scale to the right hand side. The scale is now in degrees C. The caption has been amended to state that the temperature varies around a baseline value of 18 degrees C, as the building was heated. The updated figure is shown below.

[Figure]

**Fig. 1.**

---

## Referee Comment (RC2) · C. Finlay (Referee) · 3 Aug 2018

This manuscript presents details of a simple, low cost, magnetometer system suitable both for outreach activities in schools and for scientific monitoring of large amplitude geomagnetic disturbance events.

A clear description is given of the measurement system and examples of data collected from a number of locations across the UK are presented. An illustration of the scientific value of the data in monitoring the impact of the September 2017 geomagnetic storm is given, along with a very nice movie that is available through you tube.

The authors are to be commended for conceiving and carrying out this inspiring project! It provides a splendid example of how the public can be actively involved in real-world

geoscience. Furthermore the pitfalls and challenges inherent in such a project are clearly set out; this will be of great aid to those contemplating similar activities. The manuscript is both well written and appropriately referenced.

I recommend it be accepted once the minor points/suggestions listed below are addressed.

—

1. Perhaps it may be useful to state the approx precision of the system and price (1.5 nT and 150 GBP?) of the system already in the abstract. This information is likely of great interest to many readers.

2. Section 2: Similar to a bicycle dynamo: please add a note that unlike a bicycle dynamo there is no permanent magnet in the core involved in generating the Earth's magnetic field, since the temperatures there are too high.

3. Section 2: 'the speed and density of the solar wind ... increases and energy is passed' -> 'the speed and density of the solar wind ... increases, and the interplanetary magnetic field is perturbed, resulting in energy being passed'

4. Section 2: 'generating large magnetic fields' -> generating relatively large magnetic fields (amplitudes up to 100-1000 nT)

5. Section 2.1 ' The current is directly proportional to' -> 'the current needed to produce saturation is directly proportional to'

6. Section 3: The magnetometer system consists of -> The Raspberry Pi school magnetometer system consists of

7. Section 3. 'true accuracy' Do you mean precision? Is this not a variometer without absolute measurement accuracy?

8. Section 3.1/ Fig 2. There is an offset in H between Rpi and GDAS1 early on 12th Sept of > 50 nT, any idea of the reason for this?
9. Section 3.1 'remove much of the error by backing-out the measured temperature through calibration. Has this been done? If so it might be nice to show the corrected version in Fig 3?

10. Section 4 'cover the UK in both latitude and longitude'. Perhaps a school in N. Ireland might also be of interest?

11. Section 5 'the aurora moves as far south as LER' Do you mean the magnetic signature of the aurora or the aurora as seen in visual observations?

—

---

## Referee Comment (RC3) · P. Coïsson (Referee) · 7 Aug 2018

The article "Building a Raspberry Pi School Magnetometer Network in the UK" presents a very interesting outreach projet to develop a network of low-cost magnetometers to deploy in schools. The details on the instruments a technological choices are described and a very interesting example of a major storm is presented providing the bridge between this citizen science and the research science.

The article is informative, well organised and pleasant to read. It is acceptable for publication after vey minor revisions.

Some comments:

In the introduction it is described how the low-cost magnetometers and acquisition

chains, can provide instruments useful for citizen science. It would be important to cite some other projects. Later in the manuscript the school seismometers are mentioned, it could be important to introduce these other outreach projects here. Also, other citizen science projects on magnetism could be cited (e.g. CrowMag)

Section 2 presents the Earth magnetic field, its sources and the traditional observatories that are deployed for scientific research. There is very little information bout the chains of magnetometers that exists, in many cases oriented for Space Weather applications. Some of them are used later in this work for the study of the September 2017 storm. It would be better to present them in this section, adding relevant references to these networks.

Since this is an educational article, I think it would be useful to include a figure that defines the magnetic components, in particular the definition of D and I are missing in the text.

Some additional details on clock synchronisation could also be added: how often the Raspberry Pi synchronise its clock to an internet server? Can these setting be optimised for the need of the project (e.g. every 10 s in between the record of samples). Which is the expected drift of the clock when the internet connection is lost?

It think it would be very important to give more information on the schools targeted by this project and the educational notes provided: were they prepared for elementary, high schools, colleges. . .?

A deeper discussion on the challenges of magnetometry could be extremely useful: traditionally magnetic observatory are located in isolated area, taking care that a sufficiently large radius around the sensor is preserved as much as possible from man-made noise. Schools clearly cannot meet this need. The focus could be put on the involvement of teachers and students on the project.

From a different point of view, these installations can be analysed to develop strategies

on geophysics data acquisition in environments where high level of man-made noise is expected. Strategies and algorithms to handle with the noise and retrieve geophysical signals could be developed.

Add the definition of the acronyms GDAS: Geomagnetic Data Acquisition System .

Add a reference to IGRF 12 generation.

Figure 1: add a) and b) to make more explicit what stated in the article text. Other elements (thermometer, ADC. . .) could also be indicated more explicitly on the figure for instance with labels and arrows.

Figure 3: add right vertical axis to show the unscaled deltaT

Figure 5: choose a different colour couple than orage/red that are hardly distinguishable on the figure.

The video of the storm is very informative and could be added to the article as a supplement, in order to guarantee its availability over long time.

---

## Short Comment (SC1) · 7 Aug 2018

Here is an updated version of Figure 3 with the temperature-corrected plot added on. The relevant values for this plot are:

Correction coefficients [in nT]

2nd-Order Linear Offset

Horizontal: -0.1563 3.8509 -1.5357

Vertical: 2.2721 12.6813 0.6281
* * *
[Figure]

**Fig. 1.**

---

## Author Comment (AC2) · 7 Aug 2018

Author's Response: Chris: Thank you for the positive comments on the work. We have taken your suggestions and provide a detailed response below.

1. Perhaps it may be useful to state the approx precision of the system and price (1.5 nT and 150 GBP?) of the system already in the abstract. This information is likely of great interest to many readers. Response: This was also commented by Reviewer 1. The manuscript has been updated to provide this information.

2. Section 2: Similar to a bicycle dynamo: please add a note that unlike a bicycle dynamo there is no permanent magnet in the core involved in generating the Earth's magnetic field, since the temperatures there are too high. Response: We have updated

the text to clarify this point. Added in a reference to Lowrie (2007).

3. Section 2: 'the speed and density of the solar wind ... increases and energy is passed' -> 'the speed and density of the solar wind ... increases, and the interplanetary magnetic field is perturbed, resulting in energy being passed' Response: Text has been modified as requested.

4. Section 2: 'generating large magnetic fields' -> generating relatively large magnetic fields (amplitudes up to 100-1000 nT) Response: Text has been modified.

5. Section 2.1 ' The current is directly proportional to' -> 'the current needed to produce saturation is directly proportional to' Response: Text has been modified "The current difference is directly" as this was also noted by Reviewer 1.

6. Section 3: The magnetometer system consists of -> The Raspberry Pi school magnetometer system consists of Response: Text has been changed

7. Section 3. 'true accuracy' Do you mean precision? Is this not a variometer without absolute measurement accuracy? Response: Text has been changed to say precision. The relative accuracy over a few hours is good, but it is not a full field absolute instrument so, yes, the accuracy at any given time would be poor.

8. Section 3.1/ Fig 2. There is an offset in H between Rpi and GDAS1 early on 12th Sept of > 50 nT, any idea of the reason for this? Response: I'm not entirely sure. Most likely it is some manmade background that disappeared by the middle of the day (like on the 17th Sep) as the system was in an office in Murchison House. It may in part also be due to longer term temperature drift – there is a mean value subtracted from the Rpi data rather than a linear fit over the five days.

9. Section 3.1 'remove much of the error by backing-out the measured temperature through calibration. Has this been done? If so it might be nice to show the corrected version in Fig 3? Response: Yes, I have played around with the temperature correction and made code and example data available in Python for the schools themselves to
have a go. I used a least squares fit to a second-order polynomial so get an offset (near zero), linear and second-order terms. The coefficients are quite dependent on the time-series selected though as the instrument response is non-linear with temperature. Hence, the best strategy in reality is to keep the temperature stable. For the data used in Figure 6, the temperature changes were 'backed out' as best as possible (though not fully in BGS03 for example at the end of the time-series), and the data from the BGS variometers were aligned to true north, again as best as possible (given the lack of orientation information).

10. Section 4 'cover the UK in both latitude and longitude'. Perhaps a school in N. Ireland might also be of interest? Response: Although I didn't try too hard, I didn't meet anyone from a school in NI at any of the IOP fairs or come across one through other contacts.

11. Section 5 'the aurora moves as far south as LER' Do you mean the magnetic signature of the aurora or the aurora as seen in visual observations? Response: The aurora were visible across the UK (and photographed by the BGS Aurora Camera in ESK though it was raining in LER) but I've amended the sentence to be clearer.

---

## Author Comment (AC3) · 7 Aug 2018

Author's Response: Pierdavide: We thank you very much for reading through the manuscript and providing suggestions for improving it.

Some comments: In the introduction it is described how the low-cost magnetometers and acquisition chains, can provide instruments useful for citizen science. It would be important to cite some other projects. Later in the manuscript the school seismometers are mentioned, it could be important to introduce these other outreach projects here. Also, other citizen science projects on magnetism could be cited (e.g. CrowdMag)

Response: Correct - there are a number of geophysical networks along the lines of this project particularly for citizen scientists and schools. We have amended the first

paragraph as follows: "Over the past decade, the introduction of cheap reliable computing hardware and sensors, along with the ubiquity of high-speed Internet connections mean that it is now possible create low-cost open-science networks of geophysical sensors. This has encouraged the development of data-intensive networks, for example in climate studies (Weather Observation Website: wow.metoffice.gov.uk/), seismology (BGS School Seismology Project: www.bgs.ac.uk/schoolseismology/) or cosmic ray research (the TimPix Project: www.researchinschools.org/TIMPIX) where spatial and temporal gaps in the professional scientific networks may be filled or augmented. Some networks employ existing platforms such as mobile phones using in-built sensors to record data (e.g. CrowdMag, see www.ngdc.noaa.gov/geomag/crowdmag.shtml) while others create bespoke hardware with higher accuracy than general purpose systems. In geomagnetism, high-quality low-cost fluxgate sensors have become more widely available to allow accurate monitoring of the variation of the Earth's magnetic field over time ranges of several minutes to hours."

Section 2 presents the Earth magnetic field, its sources and the traditional observatories that are deployed for scientific research. There is very little information about the chains of magnetometers that exists, in many cases oriented for Space Weather applications. Some of them are used later in this work for the study of the September 2017 storm. It would be better to present them in this section, adding relevant references to these networks. Response: I think that is perhaps too much detail for a general audience. We do note that "The global scientific magnetic observatory network fulfils this role. However, there are presently only around 200 magnetic observatories worldwide, with a very uneven spatial coverage biased towards the northern hemisphere, and Europe in particular Love (2013). Data are freely available from most of these observatories at the INTERMAGNET website (www.intermagnet.org)." So someone can look at that reference for more information or any other the other references like INTERMAGNET.

Since this is an educational article, I think it would be useful to include a figure that

defines the magnetic components, in particular the definition of D and I are missing in the text.

Response: Again, I think people interested in the detail can look up that information for themselves in Lowrie (2007) or Reda et al. (2011), for example or online.

Some additional details on clock synchronisation could also be added: how often the Raspberry Pi synchronise its clock to an internet server? Can these setting be optimised for the need of the project (e.g. every 10 s in between the record of samples).Which is the expected drift of the clock when the internet connection is lost?

Response: The drift of the clocks is a few seconds per day in a random direction. In a test of 10 systems simultaneously running without a network connection for a week, none varied by more than plus or minus twenty seconds. A Raspberry Pi with a network connection checks a Network Time Protocol (NTP) server every two minutes as part of the background scheduling. Thus, even if the connection is lost for a couple of days, the time-stamp of the retrieved data will not be too far out. However, if the network connection is lost the data are placed into a file with '.bad' in the name to differentiate them from data collected when the Rpi was correctly synchronised. The text has been updated to read: "The Raspberry Pi is not fitted with a real-time clock and therefore it is important that it is connected to the internet in order to obtain and keep the correct time. The clock drift is on the order of several seconds per day, so a Network Time Protocol (NTP) server is polled every two minutes as part of the computer's routine job schedule. An internet connection is also required for the transfer of magnetic field data onto the web."

It think it would be very important to give more information on the schools targeted by this project and the educational notes provided: were they prepared for elementary, high schools, college . . . ? Response: Table 1 has been updated to denote this. It's a mixture of primary, secondary and university – mainly secondary (11-18 year old students).

A deeper discussion on the challenges of magnetometry could be extremely useful: traditionally magnetic observatory are located in isolated area, taking care that a sufficiently large radius around the sensor is preserved as much as possible from man-made noise. Schools clearly cannot meet this need. The focus could be put on the involvement of teachers and students on the project. Response: A good point and we do note the reasons why this is only partially successful. In the UK, teachers are constantly being barraged with new requirements for planning and improving lessons which takes up much of their time. Unless they are very interested in the project then their attention will wander after a few months. It is a trade-off of access to quiet sites within a (non-ideal) school environment and visibility to the students and teacher. While we (as scientists) can advise what to do, if the school cannot achieve the quality we desire, there is not much we can really do. The project is a public outreach one as much as a scientific experiment, so we have to be flexible and fit in with the school's capabilities.

From a different point of view, these installations can be analysed to develop strategies on geophysics data acquisition in environments where high level of man-made noise is expected. Strategies and algorithms to handle with the noise and retrieve geophysical signals could be developed. Response: Very true, I completely agree. I imagine machine learning or clever quality checking algorithms (e.g. using wavelets) could be used to discriminate between man-made or natural signals. Likewise cross-correlation with local scientific observatories would be another good check of geophysical versus manmade signals. This is an entire project in itself and beyond the scope of this article though!

Add the definition of the acronyms GDAS: Geomagnetic Data Acquisition System. Response: Added in the acronym

Add a reference to IGRF 12 generation. Response: Reference to Thebault et al (2015) added

Figure 1: add a) and b) to make more explicit what stated in the article text. Other elements (thermometer, ADC...) could also be indicated more explicitly on the figure for instance with labels and arrows. Response: We've updated the caption text with much more detail. I'm not sure adding more text to the photo will work.

Figure 3: add right vertical axis to show the unscaled deltaT. Response: As suggested by Reviewer 1 too, we have changed the graph and moved the axis to the right.

Figure 5: choose a different colour couple than orange/red that are hardly distinguishable on the figure. The video of the storm is very informative and could be added to the article as a supplement, in order to guarantee its availability over long time. Response: There's a wide variety of magnetic field changes over the course of the storm ($\pm 1000$ nT) and I experimented with different colour scale limits. This version captures the essence of the electrojet moving south and north over time in the X component. I can use a different snapshot or place the other versions online if required?

---

## Referee Comment (RC4) · M. Nair (Referee) · 10 Aug 2018

Availability of low-cost electronics and spread of internet raise the possibility of filling up the spatial gaps in the environmental sensing. Add citizen science to this mix, and we get an exciting new way to collect data and to engage the public in science experiments at the same time. The authors describe development, testing and validation of a low-cost, easy-to-use vector magnetometer. They then deploy the magnetometer in 10 schools in the UK. They discuss the data collected during a geomagnetic storm and the lessons learned while working with the teachers and students. This paper is written clearly and it is easy to understand. I recommend publication of this paper in the "Geoscience Communication". I have only a few minor suggestions/questions on this paper.

[Figure]

Page 3: 1-5: Authors may also want to mention the geomagnetic substorm, which occurs almost on a daily basis at polar regions. Page 5 Lines 5-10: Where did you place the magnetometer during your initial measurements? Do you have a recommendation for keeping the magnetometer in residential/school environments? Page 5 Lines 25-30: You mention that the updated magnetometer also collects temperature data. Are you using the temperature data to calibrate the fluxgate outputs? It would be great if you can write a few lines about this. How do you deal with power outages? Can the system work off a battery? Page 6 Lines 10:15. How did you orient the magnetometer properly in your school deployments? Did you face challenges with students/teachers misaligning the magnetometers? Page 6 Lines 15:20. Can the students access the data locally? Page 7 Lines 10:15. Regarding the quiet-day signal removal. Did you use a model to remove the Sq variations? How was the long-term performance of the magnetometer systems? Did you face issues with sensor/components going bad? Page 8 Lessons Learned. You mentioned the challenges encountered during the long-term deployment in the school. A part of the issue is that the teachers have very little time to devote to this experiment. This is a common problem faced by many citizen science projects. Is paying teachers/students is an option ?

Please also note the supplement to this comment:
https://www.geosci-commun-discuss.net/gc-2018-10/gc-2018-10-RC4-supplement.pdf

---

## Author Comment (AC4) · 10 Aug 2018

Response: Manoj – we appreciate your time and efforts to read and comment on the manuscript. Please find our response to your queries and suggestions below.

Page 3: 1-5: Authors may also want to mention the geomagnetic substorm, which occurs almost on a daily basis at polar regions.

Response: We have added: 'On days without obvious magnetic activity, the solar wind loads the magnetosphere with energy over the course of several hours. This causes small 'substorms' to form in the polar regions through a process called the Dungey cycle (Dungey, 1961).'

Page 5 Lines 5-10: Where did you place the magnetometer during your initial measurements? Do you have a recommendation for keeping the magnetometer in residential/school environments?

Response: During the initial measurements, the magnetometer was placed in an unused office on the fifth floor at the south end of the former BGS building (Murchison House) far away from the goods and passenger lifts. It was covered in insulation and placed in a box to reduce temperature changes. We did recommend putting the system at the back of a classroom, away from time-varying magnetic sources (doors, metal cupboards, radiators or beside electrical ducting). The best location was in Oundle who put theirs in an equipment lab in the centre of the building. The temperature is very stable and the system is undisturbed in general. However, we can only make recommendations not dictate where the system goes.

Page 5 Lines 25-30: You mention that the updated magnetometer also collects temperature data. Are you using the temperature data to calibrate the fluxgate outputs? It would be great if you can write a few lines about this. How do you deal with power outages? Can the system work off a battery?

Response: I have produced an updated version of Figure 3 for Reviewer 2 (C. Finlay) which shows how the temperature can be corrected for. I wouldn't say it is calibrated in the usual sense as the data are not absolute, but the temperature variation can be removed. We have added in: "However, though the systems are very temperature-dependent, it is possible to remove much of the error by 'backing-out' the measured temperature variation through calibration. We demonstrate this using a second-order polynomial model to compute the least-squares best-fit coefficients between the magnetometer differences in each component (H and Z) and temperature variation. These model fits are also shown in Figure 3 denoted 'corrected'. The linear coefficients are of the order of 3.8 nT/°C for the H component and 12.6 nT/°C for the Z component. Table 2 gives all six coefficients." The system is powered from the mains electricity. If the power goes out, then it stops working. We could add an in-line battery pack, but that requires more parts (charger, voltage regulator, battery) which would cost more money

– but it is eminently possible, of course.

Page 6 Lines 10:15. How did you orient the magnetometer properly in your school deployments? Did you face challenges with students/teachers misaligning the magnetometers?

Response: In half the schools, the system was personally installed and we instructed the teacher-in-charge how best to align the system by nulling the Y-component. However, the systems tend to get moved every few months in which case the alignment has changed. The only way to deal with this is post-process the data by rotating the horizontal magnetic values back to their approximate model values or nulling the Y component using quiet time values. This is not ideal, but makes the data usable for analysis.

Page 6 Lines 15:20. Can the students access the data locally?

Response: Yes, they can either get the data from the AuroraWatch website or use a USB memory stick to collect the data from the Raspberry Pi directly. I've added a sentence to clarify that.

Page 7 Lines 10:15. Regarding the quiet-day signal removal. Did you use a model to remove the Sq variations? How was the long-term performance of the magnetometer systems? Did you face issues with sensor/components going bad?

Response: For the analysis of the September storm, the quiet time value was assumed to be the average of the magnetic field components between 02:00 and 03:00 on the 7th September prior to the storm. We did not remove the Sq variations from the data, though they are relatively small compared to the storm signal (20 nT versus >200 nT). Thus far, there have been no electronic or electrical issues with the computer, digitiser or sensor themselves. They are surprisingly robust in that sense!

Page 8 Lessons Learned. You mentioned the challenges encountered during the long-term deployment in the school. A part of the issue is that the teachers have very little

Interactive
comment
time to devote to this experiment. This is a common problem faced by many citizen science projects. Is paying teachers/students is an option?

Response: It's an interesting proposal but I don't think we'd get money under an educational or outreach grant from the UK funding bodies. As I understand it, other organisations like UK Met Office have a huge number of unpaid volunteers who send in weather readings for free, for example. There's usually plenty of interested people – it is finding them and providing them with suitable equipment and training that is the main issue.

---

## Short Comment (SC2) · 31 Aug 2018

I found the article to be very relevant in terms of the establishment of a network of low cost magnetometers. The authors showed good insight regarding the geomagnetic field, natural short term magnetic field variations and magnetometry in general. The temperature compensation that was done is very important and the alignment of each magnetometer in the network was done well. As mentioned in the article it is always required, sometimes with difficulty, to educate the public in terms of man-made magnetic noise and to find magnetically clean areas for the installation of these magnetometers.

---

## Author Comment (AC5) · 31 Aug 2018

Thank you to Dr Gouws for reviewing the manuscript. We agree that it is difficult to provide a balance between educational access to the instrument within a school environment and reducing man-made noise in the measurements.

---

## Referee Comment (RC5) · C. Webster (Referee) · 3 Sep 2018

I enjoyed reading the paper and feel it is important to share the use of low cost instruments such as the Raspberry Pi with students and researchers globally. Perhaps science centres and museums would also be willing to host a few of these instruments. The science presented in this paper is sound, well-structured, and acceptable for publication after minor revisions. It may be good to indicate the rough cost of building one of these magnetometers.

Below are a few minor comments on the text:

page 2 line 29 - seasonal* dependence or seasonally dependent*

Page 3 line 12 - The geomagnetic field can be measured with an instrument generically

(I would delete "generically" as it is not really necessary) known as a magnetometer. There are several types of instruments that can be used to* make a measurement...

page 3 line 33 - causing large electrical currents to flow. Flow where?

Page 5 line 7 – the brackets need to be closed (H...

Page 7 line 28 – use a dash between the time for consistency

Page 8 line 19 - there* is no . . .I think you mean there not here.

Spell out all acronyms such as GDASI

Where possible try not to end sentences with prepositions such as 'to' or 'in' etc.

---

## Editor Comment (EC2) · M. Joubert (Editor) · 3 Sep 2018

I would like to thank all the reviewers and other readers who contributed and commented on this article. This open review process will close on 6 September (in 3 days). I look forward to the final comments and updated version of the article from the authors.

---

## Author Comment (AC6) · 5 Sep 2018

I enjoyed reading the paper and feel it is important to share the use of low cost instruments such as the Raspberry Pi with students and researchers globally. Perhaps science centres and museums would also be willing to host a few of these instruments. The science presented in this paper is sound, well-structured, and acceptable for publication after minor revisions. It may be good to indicate the rough cost of building one of these magnetometers.

Response: Thank you for the review. The plans are freely available to the public through me and are easy enough to follow with some soldering skills, so are open to science centres for example should they be interested. The cost I have added to the

text on the recommendation of yourself and the other reviews. It costs around 150GBP at 2018 prices for all the parts.

Below are a few minor comments on the text: page 2 line 29 - seasonal* dependence or seasonally dependent* Response: corrected

Page 3 line 12 - The geomagnetic field can be measured with an instrument generically (I would delete "generically" as it is not really necessary) known as a magnetometer. There are several types of instruments that can be used to* make a measurement...

Response: OK – sentences have been modified

page 3 line 33 - causing large electrical currents to flow. Flow where? Response: added 'in the upper atmosphere'

Page 5 line 7 – the brackets need to be closed (H... Response: brackets closed

Page 7 line 28 – use a dash between the time for consistency Response: added a dash 12:00—18:00

Page 8 line 19 - there* is no Response: Changed to 'there are no easy ways to ...'

Spell out all acronyms such as GDAS Response: added in (Geomagnetic Digital Acquisition System or GDAS). This is an in-house BGS system.

Where possible try not to end sentences with prepositions such as 'to' or 'in' etc Response: I have changed the sentences where I found this.

———————————